# Single cell RNA analysis uncovers the cell differentiation and functionalization for air breathing of frog lung
Liming Chang [1,2], Qiheng Chen[1,2], Bin Wang[1], Jiongyu Liu[1], Meihua Zhang[1], Wei Zhu [1,2] ✉ & Jianping Jiang [1,2] ✉

The evolution and development of vertebrate lungs have been widely studied due to their significance in terrestrial adaptation. Amphibians possess the most primitive lungs among tetrapods, underscoring their evolutionary importance in bridging the transition from aquatic to terrestrial life. However, the intricate process of cell differentiation during amphibian lung development remains poorly understood. Using single-cell RNA sequencing, we identify 13 cell types in the developing lungs of a land-dwelling frog (*Microhyla fissipes*). We elucidate the differentiation trajectories and mechanisms of mesenchymal cells, identifying five cell fates and their respective driver genes. Using temporal dynamics analyses, we reveal the gene expression switches of epithelial cells, which facilitate air breathing during metamorphosis. Furthermore, by integrating the published data from another amphibian and two terrestrial mammals, we illuminate both conserved and divergent cellular repertoires during the evolution of tetrapod lungs. These findings uncover the frog lung cell differentiation trajectories and functionalization for breathing in air and provide valuable insights into the cell-type evolution of vertebrate lungs.

The water-to-land transition is a milestone in the evolution history of vertebrates. The emergence of air-breathing organs corresponds with the shift in oxygen medium from water to air and facilitates vertebrate terrestrialization[1,2]. In tetrapods, lungs serve as air-breathing organs, and their functionalization is a tightly regulated multistage process that involves ventilation, oxygenation, and a response to exogenous stimuli[3–5]. To execute this highly choreographed function, four major types of cells play important roles during lung development: epithelial, endothelial, mesenchymal, and immune cells. With the development of high-throughput single-cell RNA sequencing (scRNA-seq) technology, a large number of pulmonary single-cell atlases have been constructed for various organisms, such as humans[6,7], mice[8,9], non-model mammals, reptiles and birds[10], *Xenopus laevis*[11], and lungfishes[12]. These proceedings have provided a deep understanding of pulmonary cell differentiation across the vertebrate groups, spanning from water to land-dwelling species. While the cell differentiation of vertebrate lungs in evolutionary studies has become a hotspot, the dynamic processes of pulmonary cell differentiation and functionalization during vertebrate lung development are largely overlooked, particularly in the transitional groups from water to land (e.g., amphibians).

Mesenchymal cells (MSCs) occupy a distinctive position, bridging the epithelial lining and the stroma and serving as a central signaling hub that connects various cellular compartments within the lung[13]. Throughout both the embryonic and postnatal stages of lung development, signals originating from the mesenchyme play a crucial role in guiding epithelial budding, orchestrating branching morphogenesis, and facilitating subsequent structural and functional maturation[14,15]. Despite their pivotal role in development, the lineage diversity and differentiation of pulmonary MSCs remain largely elusive[16].

Alveolar epithelial cells (AECs) and endothelial cells (ECs) are essential components of the blood–gas barrier in lungs and play a vital role in gas exchange. The ultrastructure of AECs was studied in amniotes, including snakes[17], lizards[18], turtles[19], birds[20], and mammals[21,22]. It is widely supported that there are two distinct types of AEC in amniotes. The single-cell atlas of the lungs was constructed for 11 non-model species of reptiles, birds, and mammals, further confirming that there are two types of AEC in amniotes[10]. Amphibians, as a crucial vertebrate group that transitioned from aquatic to terrestrial environments, possess the most primitive lungs among tetrapods[23,24]. The presence of two types of AEC in amphibians was initially described by Okada et al. (1962) based on the relative distance from the cell

[1]CAS Key Laboratory of Mountain Ecological Restoration and Bioresource Utilization & Ecological Restoration Biodiversity Conservation Key Laboratory of Sichuan Province, Chengdu Institute of Biology, Chinese Academy of Sciences, Chengdu 610041 Sichuan, China. [2]University of Chinese Academy of Sciences, Beijing 100049, China. ✉e-mail: zhuwei@cib.ac.cn; jiangjp@cib.ac.cn

bodies to the capillaries. Goniakowska-Witalińska (1978, 1986) proposed that there is only one type of AEC, as AECs do not exhibit significant ultrastructural differences in amphibians[25,26]. Rankin et al. (2015) observed one type of AEC in the *X. laevis* base via in situ hybridization studies. They also proposed the possibility of different subtypes of AEC that may not have been identified due to the limitations of technology[27]. The cell-type evolution of ECs in amniotes has been well illustrated by Astrid et al. (2019)[28], while there is a lack of research on amphibian ECs. Recent, only one type of AEC and EC was identified in the single-cell atlases of *X. laevis* that exclusively inhabit aquatic environments throughout their life cycles[11]. However, the differentiation of AECs and ECs in amphibians is still unclear, and further scRNA sequencing studies on AECs and ECs are needed, particularly in species with distinct aquatic and terrestrial life stages.

*Microhyla fissipes* (Anura: Microhylidae) is a good model for exploring lung functionalization for air breathing during the transition from water to land, as their life cycles comprise distinct aquatic and terrestrial phases[29,30]. Our previous studies have addressed the morphological changes and molecular mechanism of lung development by using Micro-CT and whole-tissue sequencing during *M. fissipes* metamorphosis[31], while the functions of individual cells during lung development are not clear. In this study, scRNA-seq analysis was applied to further elucidate three key issues: (1) the differentiation trajectories and mechanism of MSCs; (2) the cellular and molecular processes of lung morphogenesis and functionalization during frog metamorphosis; and (3) the evolution of cell types in vertebrate lungs. This study gives a systematic view of the dynamic variations in frog pulmonary cell differentiation and functionalization during air breathing and sheds some light on the cell evolution of vertebrate lungs.

## Results
### scRNA-seq sequencing and unbiased clustering of developmental lung cells

The single cells of *M. fissipes* lungs were obtained from eight samples at four developmental stages, namely S41, S44, subadult, and adult, which are representative of the aquatic, early, middle, and late stages of terrestrial transition, respectively (Fig. 1a, b). Having passed quality control filtering (Supplementary Tables 1–4), unbiased clustering was performed on a total of 46,480 lung cells, and these cells were divided into 13 cell clusters using uniform manifold approximation and projection (UMAP, Fig. 1d, Supplementary Fig. 1).

The cell clusters were annotated based on the expression of conservative marker genes from *Xenopus* Cell Landscape and the Cell Marker 2.0 Database (Fig. 1c, Supplementary Data 1). In detail, the matrix fibroblasts (Matrix FBs) and fibroblasts (FBs) were characterized by highly expressed *LOX* and *ELN*, respectively. The chondrocytes were distinguished by their specific expression of *COL11A1*. The smooth muscle cells (SMCs) exhibited high expression levels of *MYH11* and *ACTA2*. The vascular endothelial cells (EC_Vs) were characterized by the highly expressed *CLEC14A*, while the lymphatic endothelial cells (EC_Ls) exhibited high expression levels of *CLEC4M* and *VWF*. The macrophagocytes (MACs) were characterized by the highly expressed *CTSS*. The myeloid cells were distinguished by their specific expression of *PCNA* and *HMGB2*. We classified the pulmonary epithelial cell clusters using general markers of alveolar epithelial cells (AECs) (e.g., *SFTPA2* and *SFTPC*), epithelial cells (EPIs) other than the alveolar epithelium (e.g., *CLDN4*), neuroendocrine cells (NECs) (e.g., *CHGB* and *SCG3*), ciliated cells (e.g., *CCDC39* and *CCDC173*), and ionocytes (e.g., *ATP6V1B1*).

In addition, we constructed the cellular differentiation trajectory of the developing lung based on the RNA velocity (Supplementary Fig. 2). Our analysis revealed that the MSC lineages originated from the Matrix FBs, while the other cell types exhibited independent differentiation patterns (Fig. 1e).

### Construction of differentiation trajectories of mesenchymal cell lineages

To reveal the differentiation patterns of MSCs, we extracted the MSC lineages as a subset to construct a more refined cellular trajectory (Fig. 2a,

Supplementary Fig. 3). The result indicates that the Matrix FBs exhibit the capacity for differentiation into distinct cell types, namely FBs, SMCs, and chondrocytes (Fig. 2b). We computed the terminal states and estimated the cellular fate bias of the MSC lineages (Fig. 2c, d, Supplementary Fig. 3d). Five cell fates of MSC lineages were identified, including Matrix FB fate 1, Matrix FB fate 2, FBs, SMCs, and chondrocytes, of which Matrix FB fate 2 and the FBs showed the lowest and highest fate probabilities, respectively, while Matrix FB fate 1, the SMCs, and the chondrocytes showed comparable fate probabilities (Fig. 2c, d).

In addition, driver genes have been identified for five distinct cell fates (Supplementary Data 1). Ribosomal protein subunit-related genes drive the differentiation of matrix fibroblast fate 1, including *RPS12*, *RPL32*, *RPS23*, *PRL20*, and *RPL23* (Fig. 2e). The driver genes associated with Matrix FB fate 2 are involved in the extracellular matrix structure (e.g., *VCAN*, *COL12A1*, *FBN2*, and *FBLN1*), regulating cell growth, migration, and differentiation (e.g., *MRXA5*, *TAGLN*, *MEIS2*, *PTHLH*, *TBX4*, *SOX9*, and *SNAIL2*) (Fig. 2f). Moreover, the driver genes of FBs are involved in the synthesis of collagen and extracellular matrix components (e.g., *ADAMTS2*, *COL6A3*, *COL4A2*, *TNXB*, and *MFAP5*), as well as the maintenance of tissue structure and stability (e.g., *LOX*, *MTN2*, *ANXA1*, and *FSTL1*) (Fig. 2e, f). The driver genes of SMCs play crucial roles in muscle contraction (e.g., *MYH11*, *ACTA2*, *TAGLN*, and *MYH16*), the maintenance of cell morphology (e.g., *LMOD1*, *MYH9*, and *FLNA*), and the regulation of smooth muscle cell differentiation (e.g., *ID4* and *MYOCD*) (Fig. 2g). Finally, the genes involved in the *WNT* signaling pathway (e.g., *WNT11*, *WIF1*, *WNT5A*, and *WNT16*), *BMP3*, *OSTN*, and two critical transcription factors (*OSR2-B* and *FOXF1-B*) are speculated to drive the differentiation of chondrocytes (Fig. 2g).

### Heterogeneity and temporal molecular dynamics of pulmonary epithelial cells

We identified five distinct types of epithelial cell, namely AECs, EPIs, NECs, ciliated cells, and ionocytes (Fig. 3a). The enrichment analysis of marker genes for these cell types revealed common functional attributes among certain groups (Fig. 3b). The AECs, EPIs, and ionocytes exhibited a response to hypoxia; the AECs and EPIs function on respiratory gaseous exchange. The AECs, NECs, and ciliated cells are all enriched with the GO term "secretory granule lumen". Based on enriched term, we also elucidated the specific functions of these epithelial cell types; the AECs, NECs, ciliated cells, and ionocytes exhibited the synthesis and secretion of pulmonary surfactants, the differentiation of epithelial cells, neuroendocrine regulation, the synthesis and assembly of cilia, and ion transmembrane transport, respectively (Fig. 3b). The ultrastructural analysis of alveoli further distinguished the AECs, characterized by lamellar bodies and dense microvilli, from the ciliated cells, which are columnar and bear cilia on their free surface (Fig. 3c).

Temporal molecular analysis highlighted key regulatory genes, such as *SOX9*, *NKX2.1*, *NKX2.4*, and *WNT7B*, which play pivotal roles in the AECs. The expression levels of *SOX9* decline during development, reaching their peak at S41, while those of *NKX2.1*, *NKX2.4*, and *WNT7B* peak at S44 (Fig. 3d). Additionally, the respiratory functional-related genes are involved in reducing the surface tension of pulmonary alveoli (e.g., *SFTPB* and *SFTPC*), conferring innate immunity (e.g., *SFTPA* and *LGALS4*), cellular redox reactions and metabolic regulation (e.g., *CAT* and *CYGB2*), and maintaining cellular homeostasis and ion transport (*FXYD2*, *CLIC3*, and *SLC22A31*). These genes exhibit consistent expression patterns across stages, with the highest expression level observed at S44 (Fig. 3e, Supplementary Fig. 4a). In the EPIs, regulatory genes like *WNT4*, *WNT10B*, *FGFR2*, and *HES4* were identified, with their expression levels peaking at S44 (Fig. 3d). Similarly, the genes involved in maintaining epithelial cell morphology (i.e., *KRT7*, *KRT19*, *EMP1*, and *CLDN4*) and cellular anti-oxidation reactions (i.e., *GSTT1*, *TXN*, *GSTA3*, and *CAT*) demonstrate peak expression levels at S44 (Fig. 3e, Supplementary Fig. 4b). In the NECs, regulatory genes like *SNAIL1*, *WNT11*, and *LMX1B.1* show the highest expression level at S44, while the expression level of *ID3* decreases during development, showing the highest level at S41 (Fig. 3d). Furthermore, neuroendocrine-related genes have the highest expression level at S44

(Fig. 3e, Supplementary Fig. 4c). The temporal molecular analysis of ionocytes revealed distinct expression patterns for regulatory genes (e.g., *SFRP2*, *IGFBP5*, *WNT4*, and *PTHLH*) and ion transmembrane transport-related genes, peaking at S41 and S44, respectively (Fig. 3d, e, Supplementary Fig. 4d).

Overall, these findings underscore the metamorphosis climax as a critical stage for pulmonary epithelial cell growth, differentiation, and functional switching, potentially facilitating the transition to air breathing.

## The peculiarity of the AECs in *M. fissipes*

We examined and compared the ultrastructures of alveoli between *M. fissipes* and mice. In the mouse alveoli, there are two types of AEC containing squamous AT1 and small-sized AT2 attached to the surface of AT1. The AT2 of the mice are filled with lamellar bodies in their cytoplasm and covered by microvilli on their surface (Fig. 4a). However, the alveoli of *M. fissipes* were composed of only one type of squamous AEC, which was filled

with lamellar in the cytoplasm and covered with dense microvilli on the cell surface (Fig. 4b). These results suggest that the AECs of *M. fissipes* have the combined structural characteristics of the mouse AT1 and AT2. Furthermore, the matured *M. fissipes* AECs simultaneously expressed the markers of mouse AT1 (*AHNAK* and *LGALS3*) and AT2 (*SFTPB*) based on the FISH and scRNA-seq data (Fig. 4c, d).

## Cell heterogeneity of pulmonary endothelial cells

We identified two distinct types of EC, namely EC_Vs and EC_Ls. These EC subtypes can be distinguished based on their unique expression patterns of specific marker genes. Specifically, the expression level of *VWF* is higher in the EC_Ls compared to the EC_Vs, whereas *MMRN1* and *CLEC4M* are specifically expressed in the EC_Ls. Conversely, *CLEC14A* exhibits a higher expression level in the EC_Vs compared to the EC_Ls (Fig. 5a). In addition, we identified 246 genes uniquely expressed in the EC_Ls and 96 genes uniquely expressed in the EC_Vs (Fig. 5b). These genes are involved in

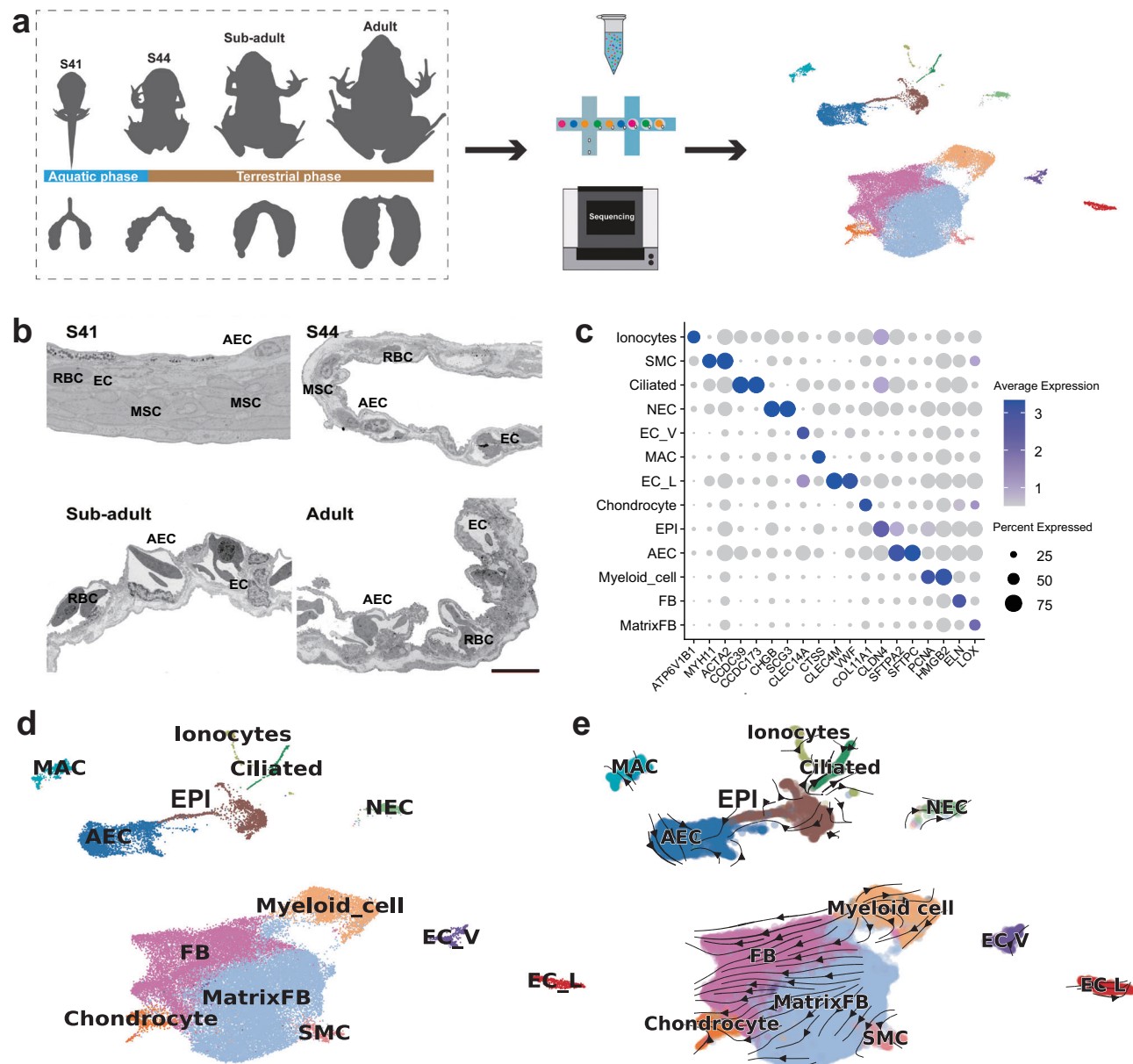

**Fig. 1 | Cellular diversity of lungs in 4 developmental stages. a** A schematic of the basic workflow for the pulmonary cell landscape in *M. fissipes* using the 10× Genomics platform. **b** Ultrastructural dynamics of alveoli showing the proportional variations in the major cell types, AEC alveolar epithelial cell, EC endothelial cell, MSC mesenchymal cell, RBC red blood cell; scale bar, 10 μm. **c** Dot plot showing the expression levels of marker genes for pulmonary cells. **d** UMAP showing unbiased clustering results of all of the filtered cells in lung. **e** UMAP visualizing the predicted pulmonary cell trajectory.

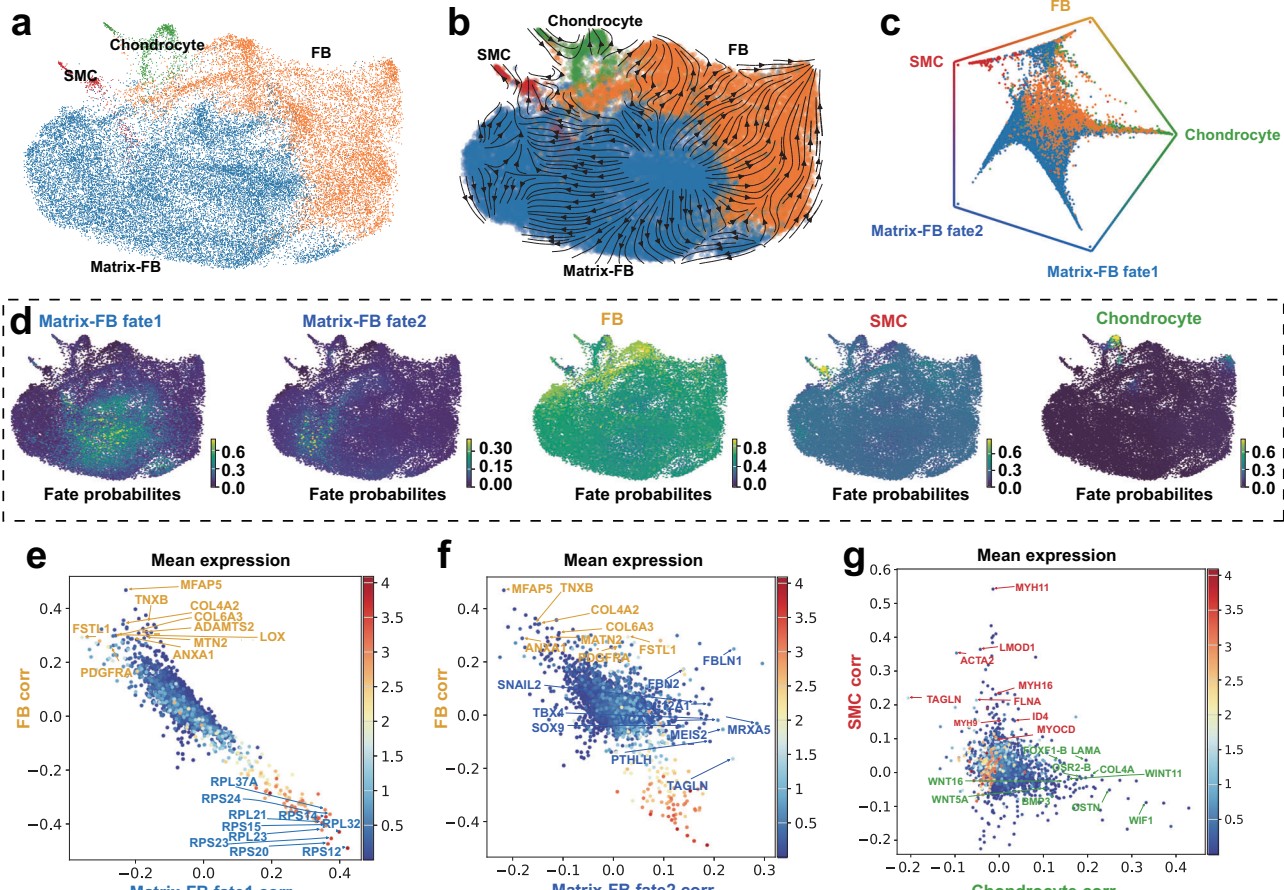

**Fig. 2 | Construction of differentiation trajectories of MSCs. a** UMAP visualization of MSCs. **b** UMAP visualizing the predicted differentiation trajectory of MSC lineages. **c** Circular plot visualizing the predicted cell fates of MSC lineages. **d** UMAP showing the cell fate probabilities of MSC lineages. **e–g** Dot plots showing the putative driver genes of different MSC fates, each dot represents a gene, colored by mean expression across all cells.

various biological processes, including the response to exogenous stimuli and cellular ion homeostasis, respectively (Fig. 5c, d). One hundred and thirty-nine marker genes shared by two types of endothelial cells were identified. These genes play crucial roles in angiogenesis, the response to wounding, and blood coagulation (Fig. 5b, e). These results suggest that the two types of EC share some common biological functions, but also exhibit functional differentiation.

## Cell-type evolution of the lungs

We integrated the scRNA-seq dataset of lungs from *M. fissipes* with the atlases of *X. laevis*, mouse, and human lungs. Cross-species analysis at the single-cell level revealed some notable findings. First, the pulmonary single-cell atlases of *M. fissipes* exhibited the strongest correlations with those of *X. laevis* (Fig. 6a). Nevertheless, some cell types, such as the EC_Ls and ciliated cells, were not identified in *X. laevis*. Second, the correspondences in cell type between *M. fissipes* and mammals are interesting (Fig. 6b, c). The AECs of *M. fissipes* lungs exhibited high similarity with both the AT1 and AT2 cells of mammalian lungs. Additionally, the EPIs of *M. fissipes* lungs showed a resemblance to the basal cells of human lungs. The EC_Vs of *M. fissipes* lungs demonstrated high similarity to the EC_Vs of mammalian lungs, although the EC_Vs of the mouse lungs displayed more specialization (e.g., EC_vacam, EC_casc, and EC_CAP) (Fig. 6d). The ciliated cells of *M. fissipes* lungs exhibited similarities to those of both the mouse and human lungs, as did the secretory cells of the human lungs (Fig. 6d). Third, the FBs and SMCs showed strong correlations in their transcriptomes across the four species. Furthermore, only two immune cell types (MACs and myeloids) were identified in the *M. fissipes* lungs, and these cells exhibited correlations with multiple types of immune cell found in the mammalian lungs (Fig. 6b, c).

Overall, these findings shed light on both the conserved and divergent aspects of lung cells across different species, providing valuable insights into the cell-type evolution of lungs.

## Discussion

In this study, we employed scRNA-seq to investigate the developmental dynamics and cellular diversity of *M. fissipes* lungs. Through comprehensive analysis across four developmental stages, we delineated the landscape of pulmonary cell types, constructed differentiation trajectories of MSCs, and revealed the temporal molecular dynamics of pulmonary epithelial cells. These findings provide insights into the evolution of cell heterogeneity in tetrapod lungs.

MSCs are the predominant major cell type during lung development. Diverse MSCs provide the architectural niche regions necessary for lung morphogenesis and functionalization. In this study, we investigated the differentiation trajectories and mechanisms of four distinct MSC types, leading to the identification of five cell fates. Matrix FB fate 1 serves as the origin of the MSC lineage, characterized by the robust expression of genes related to the ribosomal protein subunit, indicating rapid cell growth and proliferation. Matrix FB fate 2 is differentiated from Matrix FB fate 1. Among its driver genes, *TAGLN* is implicated in regulating smooth muscle contraction and cell motility[32]. *SNAIL2*[33] and *PTHLH*[34] play indispensable roles in osteoblast and bone development, respectively. *SOX9* regulates chondrocytes differentiation and pulmonary branching morphogenesis[35,36] in lungs, while *TBX4*, a key transcriptional regulator, plays an essential role in lung organogenesis[37]. Accordingly, Matrix FB fate 2 may act as a central hub for signaling, linking different cellular compartments throughout the lungs.

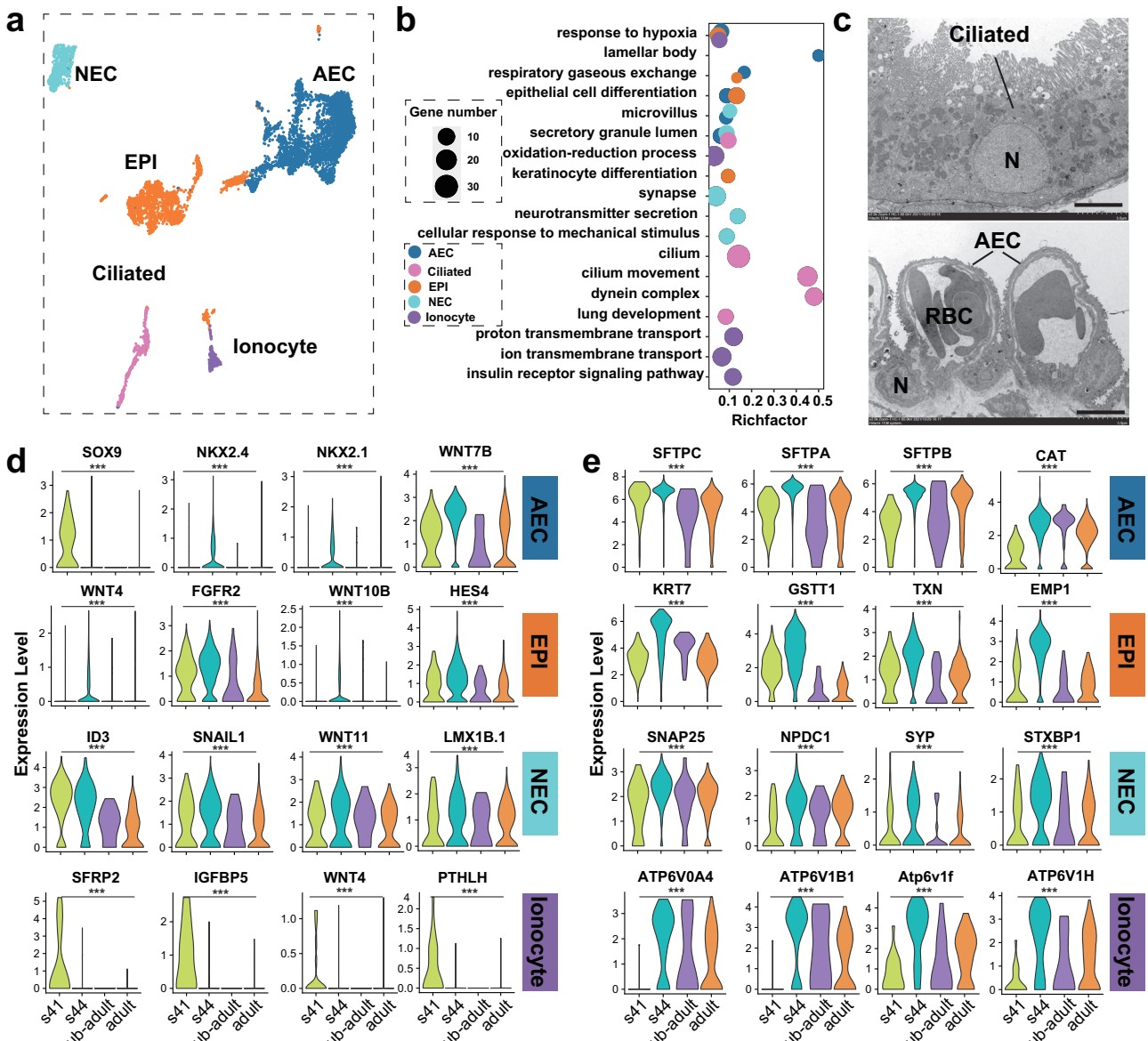

**Fig. 3 | Heterogeneity and temporal molecular dynamics of pulmonary epithelial cells in 4 developmental stages. a** UMAP visualization of pulmonary epithelial cells. **b** GO enrichment of feature genes in pulmonary epithelial cells (adjusted $p < 0.001$). **c** Ultrastructural characters of AECs and ciliated cells. Scale bar, 5 μm. Violin plots showing the temporal dynamic expression of feature genes regulating cell growth and proliferation (**d**) and genes involving in cell function (**e**) in pulmonary epithelial cells with development. (\*\*\*) indicates a significant difference among the expression level in different stages ($p < 0.001$).

The FB fate is driven by the genes involved in the synthesis of collagen and extracellular matrix (ECM) components. The ECM, derived from FBs, not only provides structural support for pulmonary cells, but also plays a critical in the regulation of developmental organogenesis and homeostasis[38]. The mature and differentiated SMC phenotype is characterized by the expression of genes encoding "contractile" proteins[39], with four contractile genes (*ACTA2*[40], *MYH11*[41], *TAGLN*[42], and *LMOD1*[43]) and their transcriptional regulator myocardin (*MYOCD*)[44,45] identified as its differentiation drivers. *MYOCD* serves as an SMC-specific transcriptional factor that interacts with the canonical single or multiple CArG boxes DNA sequence to modulate *SRF*-target contractile genes, thereby regulating the SMC differentiation processes[45]. Pulmonary chondrocytes are differentiated from Matrix FBs. It is reported *WNT5A* and *WNT11* regulate MSC chondrogenesis[46–48]. In this study, *WNT5A*, *WNT11*, *WNT16*, and *WIF1* were identified as the driver genes of chondrocytes, implicating the involvement of complex *WNT* signaling pathways in regulating chondrocyte differentiation in the lungs. The additional driver genes of chondrocytes

include *OSTN*, *OSR2-B*, and *BMP3*, with *OSTN* acting as a hormone to activate natriuretic peptide receptor *NPR3/NPR-C*, and thus promoting bone growth[49,50]. *OSR2* can regulate *SOX9A* and *COL2A1* expression in chondrocyte development[51]. Collectively, the *WNT* signaling and *BMP* signaling pathways may coactivate pulmonary chondrocyte differentiation.

This study elucidates the differentiation trajectories and mechanisms of MSCs during lung development, providing valuable insights into lung morphogenesis and functionalization. The identified regulatory genes and signaling pathways offer potential targets for future investigations aimed at modulating lung development processes.

We elucidate the temporal molecular dynamics of pulmonary epithelial cells (i.e., AECs, EPIs, NECs, and Ionocytes) during metamorphosis, aiming to comprehend the pivotal process of air breathing adaptation. In the AECs, we observe a transcriptional peak of *SOX9* at S41. *SOX9* is involved in the lung epithelium during branching morphogenesis by balancing cell proliferation and differentiation, as well as regulating the extracellular matrix[36,52]. Its expression pattern in the lungs of *M. fissipes* suggests robust

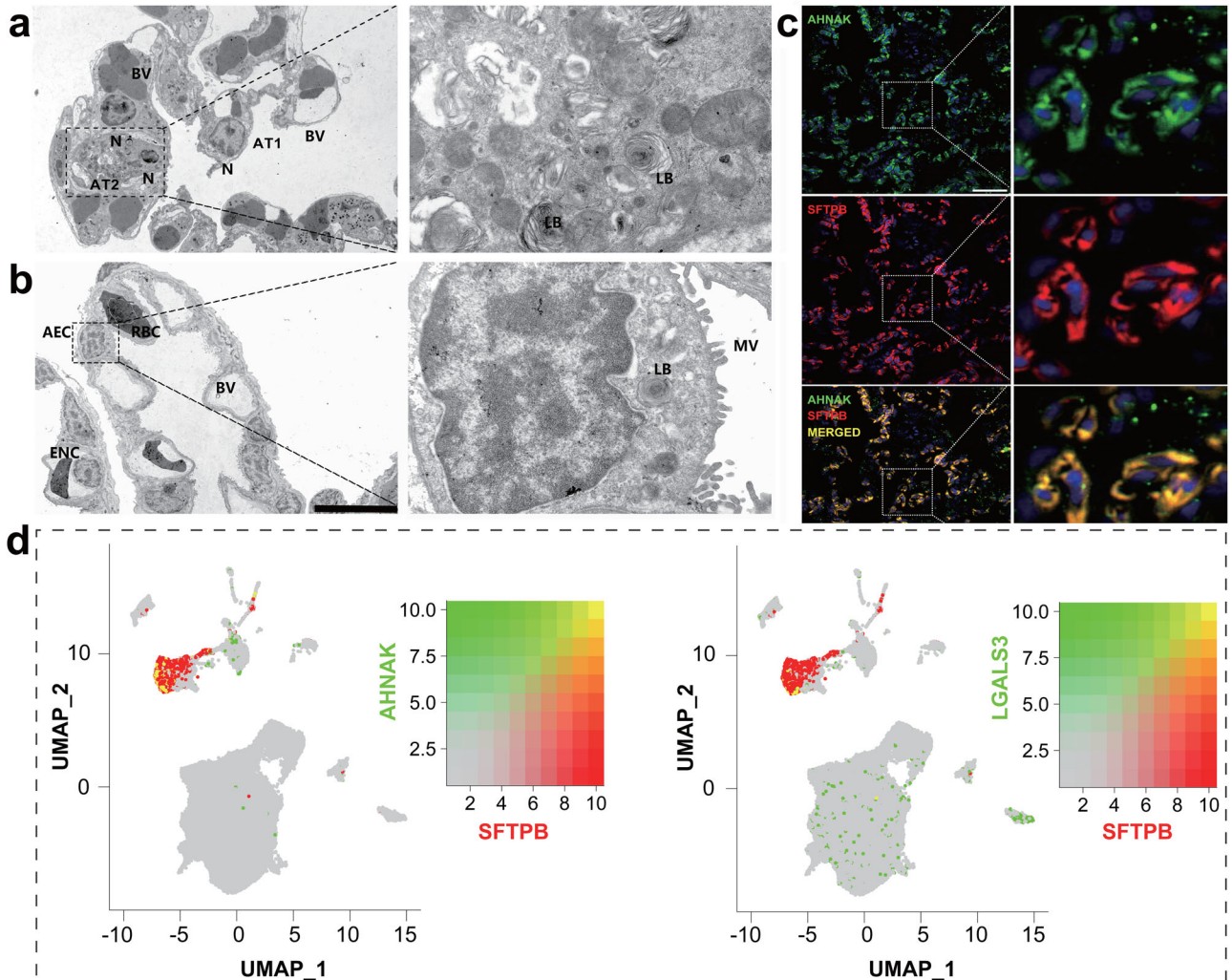

**Fig. 4 | The peculiarity of AECs in adult *M. fissipes*.** Ultrastructural characters of alveoli in adult mouse (**a**) and *M. fissipes* (**b**). BV blood vessel, LB lamellar body, MV microvillus, N nucleus, scale bar, 10 μm. **c** FISH presenting the expression of *AHNAK* (green) and *SFTPB* (red) in AECs of *M. fissipes*, co-expression showed yellow fluorescence. Scale bar, 50 μm. **d** The UMAP plots presenting co-expression of *AHNAK* and *LGALS3* (green) with *SFTPB* (red) in AECs of *M. fissipes* based on scRNA-seq data.

pulmonary morphogenesis at this development stage. This is unlike *SOX9*, *WNT7B* and *NKX2.1*, which reach their transcriptional peak at S44, the critical stage for water-to-land transition. It is reported that *WNT7B* is the only member of the *WNT* family of autocrine/paracrine signaling molecules whose expression is restricted to the airway epithelium during embryonic lung development, while *NKX2.1* can bind to a specific subset of consensus DNA *cis*-elements within the *WNT7B* promoter and activate the expression of *WNT7B*[53], with a role in coordinating pulmonary epithelial cell differentiation and regulating pulmonary surfactant homeostasis[54,55]. Similar to *WNT7B* and *NKX2.1*, the expression levels of pulmonary surfactant-associated proteins (*SFTPA*, *SFTPB*, and *SFTPC*) also peak at S44. These *SFTPs* are fundamental in the maintenance of breathing by reducing the surface tension of pulmonary alveoli and conferring innate immunity[56,57]. Moreover, the genes whose expression peaks at stage 44 additionally include those involved in cellular redox reactions and metabolic regulation, and those that maintain cellular homeostasis and ion transport exhibit the highest expression levels at this development stage. These findings indicated that S44 is associated with a burst in transcriptions of genes associated with gas exchange. Similar to AECs, a cluster of genes exhibited temporary transcriptional upregulation in EPIs at S44. These included regulatory genes like *WNT4* and *FGFR2*, which play significant roles in lung development and branching morphogenesis[58], as well as cellular functional genes involved in maintaining epithelial cell morphology and antioxidant reactions. In the

NECs, the expression of *ID3*, a TF regulating the cell cycle and the survival of neural crest progenitors[49], exhibited the highest level at S41, while the expression levels of *WNT11* and *LMX1B.1*, the key regulator of neurogenesis[59–61] and neuroendocrine-related genes, peaked at S44. Finally, ionocytes demonstrate highest expression levels of regulatory genes and functional genes (e.g., ion transmembrane transport-related genes) at S41 and S44, respectively. These temporal molecular dynamics across pulmonary epithelial cell types reveal consensus patterns; the genes governing cell differentiation and lung morphogenesis normally are typically triggered at or before stages S41, while the genes operating metabolic processes and maintaining homeostasis are activated at S44. These findings align with the results of the previous studies and highlight the profound changes in the lungs at cellular and molecular levels during the metamorphic climax, marking their functional maturation.

Our investigation suggested the involvement of the WNT signaling pathway in regulating the differentiation of multiple pulmonary epithelial cell types. Remarkably, distinct WNT genes exhibit cell-specific expression patterns in epithelial types. This nuanced expression highlights the complexity of WNT signaling in orchestrating diverse cellular functions across various pulmonary epithelial cell populations. Such specificity suggests regulatory mechanisms tailored to the unique developmental and functional requirements of each cell type within the lung tissue. Collectively, our results underscore the intricate

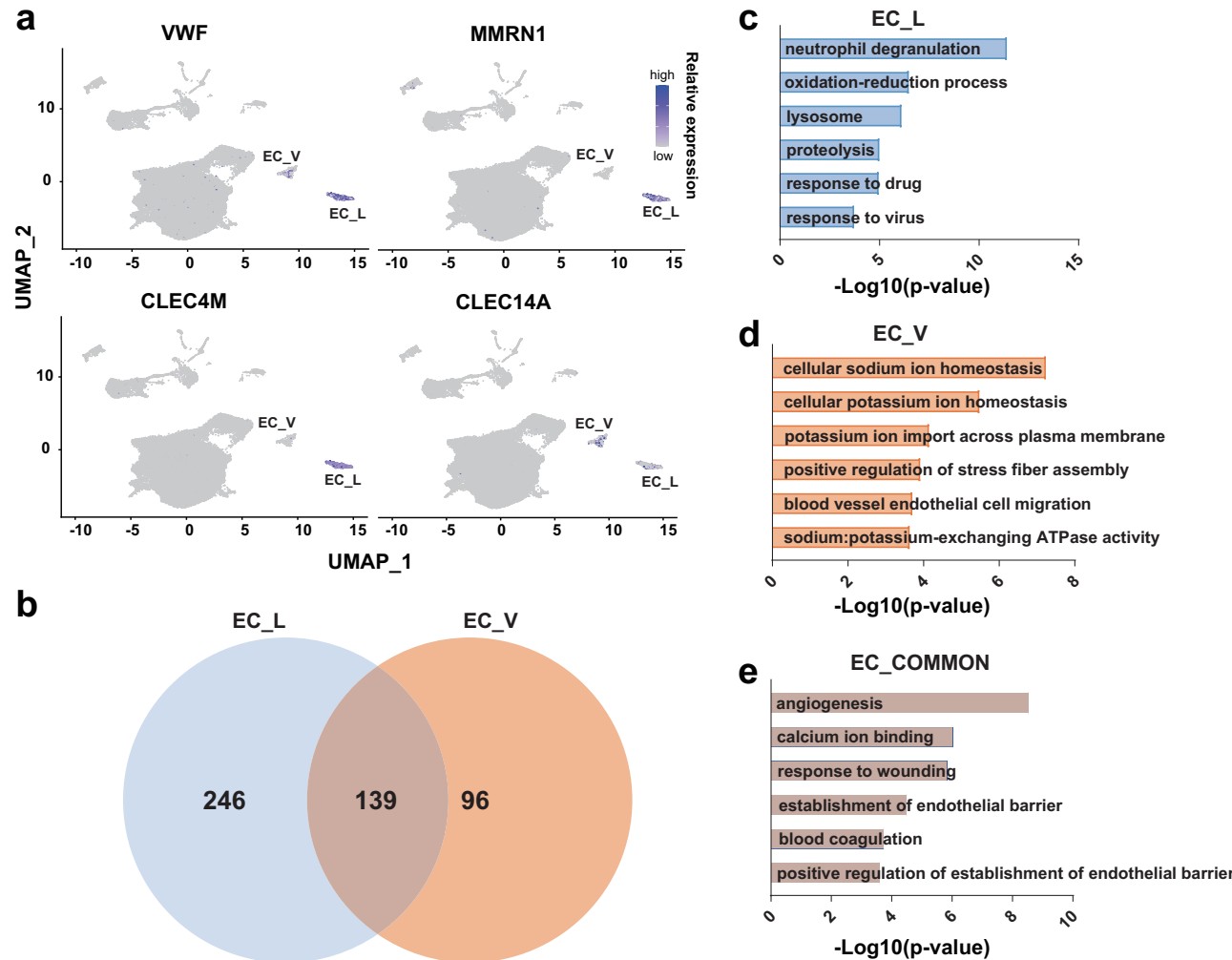

**Fig. 5 | Cell heterogeneity of pulmonary ECs. a** UMAP visualizing the feature genes of pulmonary ECs. **b** Venn diagram showing the gene number uniquely expressed and shared by two types of ECs. The blue, orange, and brown area represent the gene number uniquely expressed in EC_Ls, EC_Vs, and shared by two types of ECs, respectively. Bar plots showing the GO enrichment of feature genes uniquely expressed in EC_Ls (**c**) and EC_Vs (**d**) and shared by two types of ECs (**e**).

coordination among diverse pulmonary epithelial cell types in facilitating lung functionalization for air breathing.

It has been proposed that the evolutionary divergence of vertebrate organs is primarily driven by cell types rather than tissues[11,62,63]. We performed cross-species integration to compare the cellular repertoires of amphibian and terrestrial mammal lungs. Although adult *M. fissipes* and *X. laevis* inhabit distinct environments[11], their cell atlases exhibited strong correlations, suggesting that phylogeny is a more significant determinant of lung cellular heterogeneity than the environment. Several cell types of *M. fissipes*, such as EC_Ls and ciliated cells, had no counterpart in *X. laevis*[11]. This may be explained by the low proportions of lung cells in the single-cell atlas of *X. laevis*[11], leading to some cell types not being captured, as ciliated cells are evolutionarily conserved and have been identified in lungfish[12], *M. fissipes*, and mammal lungs[9].

Interestingly, only one type of EC was identified in the lungfish[12] and *X. laevis* lungs[11], whereas more than one type of EC was identified in the *M. fissipes* and mammal lungs[9]. Further analysis revealed that *VWF*[11], the EC maker of *X. laevis*, was commonly expressed in the EC_Ls and EC_Vs of *M. fissipes*. And the EC_Ls and EC_Vs of *M. fissipes* exhibit strong correlations with the EC_Ls and EC_Vs of the mammals, respectively, underscoring the robustness of EC types in *M. fissipes* lungs. Thus, there may be two explanations for the difference in EC types between *X. laevis* and *M. fissipes* lungs. (1) *M. fissipes* has evolved two distinct EC types to adapt to terrestrial environments due to their distinct habitats, or (2) the limited discernment

on the ECs of *X. laevis* single-cell atlases, which were constructed with multiple tissues, resulted in a relatively small number of lung cells.

The analysis of cross-species pulmonary atlases revealed that the AECs of amphibians exhibited the transcriptional characteristics of both the AT1 and AT2 cells of mammalians. In addition, the ultrastructure and FISH of alveoli proved that the AECs of amphibians have the combined structural characteristics of mouse AT1 and AT2. It is evidenced that there is also only one type of AEC in lungfishes. We speculated that the evolution of different types of AEC in amniotes may present an advanced adaptation to terrestrial life, particularly as their skin hindered respiration. The pulmonary immune cell types in the amphibian lungs exhibited correlations with multiple types of immune cells found in the mammalian lungs. This suggests that, as mammals adapt to terrestrial life, they develop more types of pulmonary immune cells to cope with complex and diverse terrestrial environments. These findings provide valuable insights into the primitive nature of air-breathing organs in amphibians, highlighting the need for future research to elucidate the evolutionary adaptations shaping pulmonary diversity across vertebrates.

## Materials and methods
### Animals
Four egg clutches (ranging from 200 to 500 eggs) of *M. fissipes* were kept in a lab and placed into 12 aquatic containers (length 42 × width 30 × depth 10 cm, water depth = 5 cm) and hatched (water temperature 25 ± 0.5 °C;

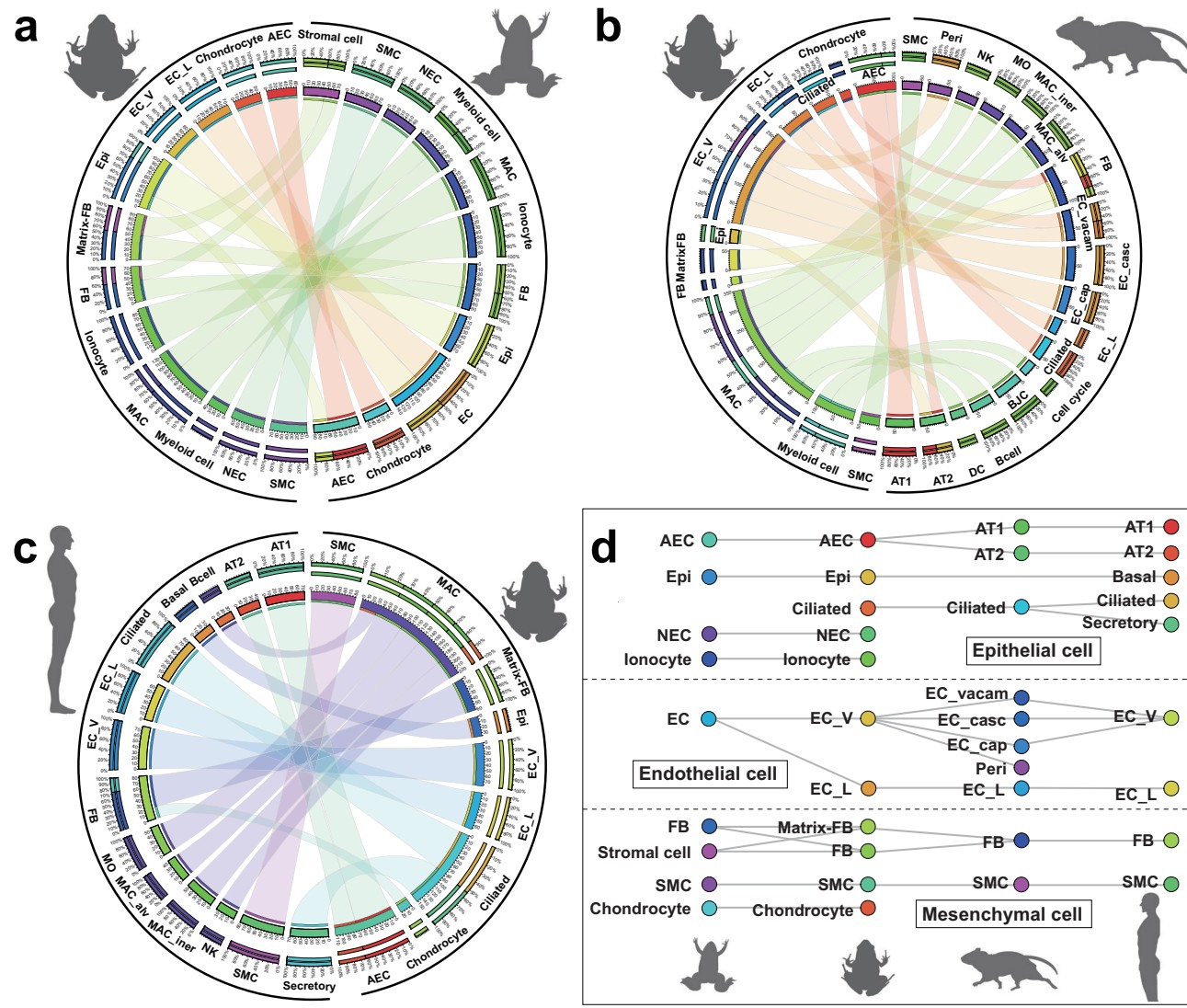

**Fig. 6 | Cell type evolution of the lung.** Circle plot showing the similarity of cell lineages between *Microhyla fissipes* and *Xenopus laevis* (**a**), mouse (**b**), and human lung (**c**), respectively. **d** Sankey plot showing the pairwise cell type similarities of the *X. laevis*, *M. fissipes*, mouse, and human lung. The different colors of dots represent various cell type. The illustrations of *X. laevis*, mouse, and human are created with MedPeer (medpeer.cn).

light/dark = 12:12 h; lights on at 7:00 h, and off at 19:00 h). The hatched tadpoles were fed a solution of boiled chicken egg yolk once a day for 2 days. The tadpoles were fed spirulina powder (China National Salt Industry Corporation) once a day, and the water was replaced every 2 days. The developmental stages of tadpoles were identified according to the staging table reported by Wang et al. (2017)[30]. The adult *M. fissipes* frogs used in this work were the parents of the tadpoles and subadults above, which were collected from farmlands (E 103.459885°, N 30.744614°, 701 m) located in Shifang City, Sichuan Province, China.

All the procedures applied for this study were approved by the Institutional Ethics Committee of Animal Ethical and Welfare Committee of Chengdu Institute of Biology, Chinese Academy of Sciences (permit: CIB20190201), and all the methods were carried out in accordance with the Code of Practice for the Care and Handling of animal guidelines. This study is reported in compliance with the ARRIVE guidelines. We have complied with all relevant ethical regulations for animal use.

**Tissue dissociation and preparation of single-cell suspensions**
For scRNA-seq, 90, 70, 30, and 5 individual lungs at stages 41, 44, sub-adult (2 months after hatching), and adult, respectively, were dissociated for the preparation of single-cell suspensions. Two biological replicates were prepared for each developmental stage. The lung tissues were placed

into a culture dish on ice with 0.7 × HBSS (calcium-free and magnesium-free) and cut into 0.5 mm² pieces, followed by being washed with 0.7 × HBSS. The lung tissues were grinded with ground glasses and transferred in a dissociation solution (0.35% collagenase IV5, 2 mg/ml papain, 120 Units/ml DNase I) on ice for 15–20 min, and then pipetted 4–5 times with a Pasteur pipette. The tissues were washed with 0.7 × HBSS and dissociated with trypsin in 34 °C water bath for 8 min. Digestion was terminated with 0.7 × HBSS containing 10% fetal bovine serum (FBS, V/V). The cell suspension was pipetted 5–10 times with a Pasteur pipette and filtered by passing through a 30 μm stacked cell strainer, and then centrifuged t at 300 g for 5 min at 4 °C. The cell pellets were resuspended in 100 μl 0.7 × HBSS (0.04% BSA). The suspension was then centrifuged at 300 g for 5 min at room temperature and resuspended in 100 μl Dead Cell Removal MicroBeads (MACS 130-090-101), followed by dead cell removal using the Miltenyi ® Dead Cell Removal Kit (MACS 130-090-101). The suspension was then resuspended in 1× HBSS (0.04% BSA) and centrifuged at 300 g for 3 min at 4 °C (repeated 2 times). The cell pellets were resuspended in 50 μl of 0.7 × PBS (0.04% BSA). The overall cell viability was confirmed using trypan blue exclusion, and the cells with greater than 85% viability were chosen for further analysis. Single-cell pellets were counted using a Countess II Automated Cell Counter and concentration adjusted to 700–1200 cells/μl.

## Chromium 10× Genomics library and sequencing

scRNA-seq libraries were prepared following the manufacturer's instructions in the Chromium Next GEM Single-Cell 3′ Reagent Kits v3.1 (https://www.10xgenomics.com/support/single-cell-gene-expression/documentation/steps/library-prep/chromium-single-cell-3-reagent-kits-user-guide-v-3-1-chemistry Chemistry) from 10× Genomics, Inc. (Pleasanton, CA). The libraries were sequenced on an Illumina NovaSeq 6000 sequencing system (paired-end multiplexing run, 150 bp) by LC-Bio Technology Co. Ltd. (Hangzhou, China).

## Pre-processing and quality control of scRNA-seq

The sequencing results were demultiplexed and converted to FASTQ format using Illumina bcl2fastq software (version 2.20). Sample demultiplexing, barcode processing, and single-cell 3'gene counting were preformed using the Cell Ranger pipeline (https://support.10xgenomics.com/single-cell-geneexpression/software/pipelines/latest/what-is-cell-ranger, version 3.1.0), and the scRNA-seq data were aligned to the *M. fissipes* reference genome. The Cell Ranger output was loaded into Seurat (version 4.0.2). High-quality single cells must pass the following quality control thresholds: a number of detected genes per cell >200 and <3000 (all the genes expressed in less than three cells were removed), and mitochondrial DNA-derived gene expression <15%.

## Integration of scRNA-seq datasets from four developmental stages

The datasets from eight sequencing libraries were integrated using an anchoring procedure implemented in Seurat v4.0.2. In details, the data were normalized with fast integration using the reciprocal PCA (RPCA) method using the "NormalizeData" function, and subsequently scaled using Pearson Residuals with a scale factor of 10,000. The top 3000 highly variable features were selected using the "SelectIntegrationFeatures" function, the integration anchors were found using the "FindIntegrationAnchors" function, and then the data were integrated using the "IntegrateData" function.

## Identification of cell clusters

Following integration, principal component analysis was performed using the "RunPCA" function with the default parameters. UMAP dimensionality reduction methods were used based on the top 50 principal components (PCs) using the "RunUMAP" functions, respectively. Moreover, the unsupervised clusters were identified by establishing the top 50 PCs and with a clustering resolution of 0.15 using "FindNeighbors" and "FindClusters" functions. To avoid any potential confounding factors, we removed the cell clusters exhibiting specific expression of red blood cell markers (i.e., *HBA* and *HBB*) from the single-cell dataset. Then we repeated the process of "identification of cell clusters" again using the filtered dataset to conduct unbiased clustering of developmental lung cells.

## Identification of differentially expressed genes (DEGs) across clusters

The FindAllMarkers function implemented in Seurat v4.0.2 was used to identify the DEGs across clusters and developmental stages with the options "min.pct = 0.25, logfc.threshold = 0.25, test.use = wilcox". Multiple test correction for the *p* value was performed using the Bonferroni method, and 0.05 was set as a threshold to define significance.

## Gene ontology (GO) enrichment analysis

Gene ontology (GO) analysis was conducted using KOBAS 3.0[64]. The Benjamini–Hochberg (BH) method was used for multiple test adjustment, and 0.05 was set as a threshold to define significance.

## Annotation of cell cluster

The cell clusters were annotated based on the expression levels of conservative cell markers from the *Xenopus* Cell Landscape[11] and Cell Marker 2.0 Database[65]. The cell markers were visualized using the "FeaturePlot" function.

## Trajectory analysis of mesenchymal cell lineage differentiation

We applied scVelo's dynamical model (release 0.2.3) to derive the trajectory analysis of mesenchymal cell lineage based on the RNA velocity using spliced and unspliced counts from Kallisto. Next, we used the related CellRank (release v2)[66] package to compute the terminal states using "g.predict_terminal_states" and estimated the cellular fate bias of the MSC lineages using "g.compute_fate_probabilities". We visualized the combined fate probabilities towards all the terminal states jointly in a circular projection. The driver gene of different cell fates were identified using "g.compute_lineage_drivers".

## Collection of orthologous genes

We used NCBI blast+ v2.9.0 to perform the BLAST analysis of the protein sequences of *Microhyla fissipes* and the other species. The genome assemblies used were human GRCh38.p12, mouse GRCm38.p6, and *X. laevis* Xenbase v9.1. We set the BLAST parameters as follows: -f 6 --sensitive --evalue 1e-6. In this study, we considered one-to-one orthologous and one-to-many paralogous pairs for further analysis.

## Cross-species analysis

We collected human, mouse[9], and *X. laevis*[11] cell–gene matrices and cell annotations of the lungs. The gene expression data for each species underwent normalization to find the sum of transcripts and perform multiplication. To compare the transcriptomes across species, we performed SAMap[67] analysis on the *M. fissipes*, *X. laevis*, mice, and humans. The cell-type pairs with mapping scores higher than 20 were plotted between species using the package Circlize. The illustrations of *X. laevis*, mouse, and human are created with MedPeer (medpeer.cn).

## Transmission electron microscopic observation

The lung tissue samples were obtained from S41, S44, subadult, and adult individuals, with three biological replicates for each stage. Fresh lung tissues were collected and cut into small blocks (1 mm³). These tissue blocks were fixed in 3% glutaraldehyde at 4 °C for 6 h, followed by rinsing in 0.1 M Sorensen's phosphate buffer (pH 7.4) three times. Subsequently, the blocks were post-fixed in 1% osmium tetroxide in the same buffer for 2 h. To proceed, the tissue blocks were dehydrated using a graded series of ethanol (30%, 50%, 70%, 80%, 95%, and 100%) for 20 min for each concentration. Afterward, the blocks were penetrated with a mixture of acetone and EMBed 812 overnight at 37 °C. The tissue blocks were then embedded in EMBed 812, and the embedded models with resin and the samples were polymerized at 65 °C for more than 48 h. Next, the resin blocks were cut into ultrathin sections measuring 60–80 nm using an ultramicrotome (Leica UC7) and Diamond slicer (Daitome Ultra 45). These ultrathin sections were placed onto 150-mesh cuprum grids with formvar film and stained in a solution of 2% uranium acetate saturated alcohol solution for 8 min. Following rinsing with 70% ethanol and ultrapure water, the ultrathin sections were stained with 2.6% lead citrate for 8 minutes. After drying with filter paper, the cuprum grids were placed into a grid board overnight at room temperature. Finally, the ultrastructure of the lungs was observed under a TEM (Hitachi, HT7800/HT7700) operating at 60 kV, and images were captured using a CCD digital camera (APTINA CMOS Sensor, San Jose, CA, USA).

## RNA fluorescence in situ hybridization (FISH) assay

The lung tissue samples were obtained from S41, S44, subadult, and adult specimens for RNA FISH. The fluorescence-conjugated probes of *SFTPB* and *AHNAK* were generated in line with the protocols of Servicebio Technologies for FISH, and probe sequences are provided in Supplementary Table 5. The detailed step-by-step RNA FISH protocol used in this study can be found in the manuals of Servicebio Technologies.

## Statistics and reproducibility

Gene different expression analysis across clusters was performed using the Wilcox test. Multiple test corrections were performed to determine

statistical significance using the *p* value (Bonferroni). The Benjamini–Hochberg (BH) method was used for multiple test adjustments for gene ontology enrichment analysis. Unless stated elsewhere, FISH and TEM were performed with at least three experimental replicates. No statistical method was used to predetermine the sample size. No data were excluded from the analyses. The filtering criteria for the low-quality cells are provided in the section above. Randomization was not performed in this study.

## Reporting summary

Further information on research design is available in the Nature Portfolio Reporting Summary linked to this article.

## Data availability

scRNA-seq data of *M. fissipes* lung are available in the GSA (CRA010691). The source data behind the graphs in the paper are provided in Supplementary Data 1. All the protein sequencing data and annotation of *M. fissipes* are provided in "Figshare [https://doi.org/10.6084/m9.figshare.25814920.v1]" and "Figshare [https://doi.org/10.6084/m9.figshare.25814923.v1]", respectively. The published scRNA-seq data of human and mouse lung used in this study are available in the GEO database under accession code GSE133747. The published scRNA-seq data of *X. laevis* were available at "Figshare [https://doi.org/10.6084/m9.figshare.19152839]"[68]. All other data are available from the corresponding author (or other sources, as applicable) on reasonable request.

## Code availability

In this study, we used software general workflow codes, without generating any new code. The R Project for Statistical Computing: R (v4.04); Preprocessing and quality control of scRNA-seq: Cell Ranger (v3.1.0); Integration of scRNAseq datasets: Seurat (v 4.0.2); Pseudotime trajectory analysis: Cellrank 2; Gene ontology (GO) enrichment analysis: KOBAS 3.0.

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

## Acknowledgements

This work was supported by National Natural Science Foundation of China (Grant no. 32300350); China Postdoctoral Science Foundation (Grant no. 2023M733438); Natural Science Foundation of Sichuan Province of China (Grant no. 2024NSFSC1182); Important Research Project of Chinese Academy of Sciences (Grant no. KJZG-EW-L13).

## Author contributions

Conceptualization: J.P.J., W.Z., and L.M.C.; methodology: J.P.J., W.Z., and L.M.C.; software: L.M.C. and W.Z.; formal analysis: L.M.C. and W.Z.; investigation: L.M.C., W.Z., Q.H.C., and B.W.; resources: L.M.C., W.Z., Q.H.C., M.H.Z., B.W., and J.Y.L; data curation: L.M.C. and W.Z.; writing—original draft preparation: L.M.C.; writing—review and editing: L.M.C., W.Z. and J.P.J.; visualization: L.M.C. and W.Z.; supervision: J.P.J.; project administration: J.P.J.; funding acquisition: J.P.J and L.M.C. All authors participated in the manuscript preparation and approved the final version of the manuscript.

## Competing interests

The authors declare no competing interests.
