## [Peer Review File · Communications Biology]

Reviewers' comments:

Reviewer #1 (Remarks to the Author):

This manuscript reports ScRNAseq analysis of lung development in *Microhyla fissipes*, a land-dwelling frog species. The ScRNAseq analysis included lungs at 4 developmental stages, namely S41, S44, Sub-adult (post metamorphosis), and Adult, representative of the aquatic stage, early, middle, and late stages of terrestrial transition, respectively. Through ScRNAseq analysis, they identified the 4 major types of lung cells, which could be further divided into 21 subtypes of cells, and found that the proportions of 4 major cell types varied significantly among the 4 stages analyzed. They further analyzed stage-associated differences of some cell types and their featured differential expression of genes and transcriptional factors (TFs), and the potential developmental linkage of the cells through differentiation trajectory analysis over pseudo time. They found that the *Microhyla fissipes* lung comprised only one type of alveolar epithelial cells (AECs), contrasting with the facts that two types of AECs in the lungs of mice and other mammals, and the AECs exhibited temporal dynamic expression of feature genes and TFs with development. This a comprehensive manuscript that revealed how frog lung likely developed through cell type transformation during metamorphosis to adopt its habitual life change from aquatic to terrestrial environment. The manuscript is highly valuable for exploring the molecular mechanism underlining lung development in vertebrates. Below are some points for consideration:

1. Through ScRNAseq analysis, only one type of AECs was identified (Fig. 1C), which expressed mouse AEC2 marker SFTP (Fig. 4B and 5B-C), along with one type of AEC progenitors (Fig. 1C), which expressed the epithelial progenitor marker gene SOX9 (Fig. 4B). How was this type of AEC progenitors compared with mouse AEC progenitor cells, e.g., any conservations among the featured differential expression of genes and/or TFs?
2. In the colocalization experiments (Fig. 4B), the FISH signal for SOX9 seemed to be very weak and need to be improved. Other than SOX9, did the authors try FISH with other epithelial progenitor marker genes to independently validate the AEC progenitor cluster?
3. In Fig. 5B, the FISH indicated that AHNK (one of the mouse AEC1 marker genes) colocalized with SFTP (mouse AEC2 marker gene) very well, except for some small green dotted signals in the merged images. What were those green dotted signals representing? In Fig. 5C, on the UMAPs, the 4 mouse AEC1 marker genes including AHNK seemed not to co-express strongly in the AEC cluster (the AEC cluster were strongly red instead of yellow as expected). What is the explanation for this discrepancy?
4. There were some inconsistencies between figures and figure legends: for example, Fig1 E and F mismatched with the corresponding figures; in Fig. 5, the figure legends regarding panels A to D didn't match the panels in the figure either (only A to C presented in the figure). The authors should also carefully check the citations of these figures in the text to make sure they were correctly referred in Results and Discussions.

5. In Fig. 3A as well as in the Supplemental Fig. 2C, 3A-B, 4B-C &4E, 5B &5D, there were colored highlight bars with different length in the GO terms panels. It is useful to define what different colors represent.

6. There were some grammar errors in the text and figure legends should be corrected. The citations to the figures in the Results and Discussion should be carefully verified to correct mistakes, such as “Four distinct types of RBCs were identified (Figure 1A).” (line 309): Fig1A had nothing to do with RBC identification.

Reviewer #2 (Remarks to the Author):

The work by Change et al. provides a comprehensive analysis of cell heterogeneity in amphibian lungs, with a specific focus on characterizing AECs (alveolar epithelial cells) and their developmental trajectories during the transition to air breathing. Leveraging the unique attributes of land-dwelling frogs, particularly *Microhyla fissipes*, the researchers performed single-cell RNA sequencing (scRNAseq) across four developmental stages to elucidate gene expression dynamics during the terrestrial transition. Through bioinformatics methods, the authors unveiled the composition and heterogeneity of cell types throughout this transition, providing a crucial resource for advancing our understanding of lung development.

The study yields novel insights into the temporal dynamics of epithelial and endothelial cells during lung development, highlighting the existence of distinct cell states. Identification of transition cell states, exhibiting both proliferative and differentiating potentials, especially during critical transitional stages, suggests a strategic mechanism for rapid morphogenesis of the blood-air barrier during the metamorphic climax. The researchers emphasize the maintenance of growth, proliferation, and differentiation capabilities in epithelial and endothelial cells, orchestrated by specific transcription factors. This agrees with the activation of adaptive responses, such as surfactant proteins and oxidoreductases, signals preparation for the transition from a hypoxic to a hyperoxic environment. Delving into an evolutionary perspective, the study proposes that the diversity observed in AEC types and progenitors in amniotes reflects an advanced adaptation to terrestrial life.

While the technical foundation of the paper is robust, the authors are cautioned against the potential overinterpretation of results. This is particularly relevant given the paper's primary focus as a resource, though the authors are encouraged to provide further elaboration on result interpretation, especially in the Discussion section. Additionally, addressing the interpretation of cell states, both the proliferative and mature transitions, and exploring factors driving these transitions at bifurcation points would enhance the paper's depth.

Minor comment: Please make sure all abbreviations are spelled out in the text.

Reviewer #3 (Remarks to the Author):

Chang et al. conducted a study on the cell atlas of the developing lung of a land-dwelling frog using single cell RNA-seq. They also performed comparative cellular analyses across different cell types and developmental stages, identifying cell types such as AECs and candidate genes (e.g., TFs) that may be involved in the lung development of land-dwelling frogs. While this manuscript presents an important resource for studying the evolution and development of vertebrate lungs, it lacks detailed analysis and robust results regarding these findings. Here are my main comments:

1. The overall cell classification is too simplistic. I suggest that the cell annotation for the amphibian lung should reference the *Xenopus* Cell Landscape from Dr. Guo Guoji's laboratory, which can be found at <https://bis.zju.edu.cn/XCL/landscape.html>.
2. It is essential to assess the reproducibility of the cell proportions in each sample. I recommend increasing sample replication or employing alternative cell counting methods for validation. Neither fig.1D nor Supplementary Figure 1C demonstrate the consistency of cell proportions among different individuals at the same developmental stage. Additionally, the proportion of red blood cells appears to be quite high. The manuscript does not mention the treatment of red blood cells (e.g., cleaning, perfusion), which could introduce errors in comparing cell proportions across different development stages.
3. The abstract and methods sections indicate the use of scRNA-seq technology, but the figure labels mention snRNA-seq. This discrepancy should be clarified.

In addition, there are many spelling and punctuation errors in the manuscript, which requires a native english speaker to revise.

Point-by-point response letter to Reviewers' comments

Manuscript number: COMMSBIO-23-3846A

Title: " Single cell RNA analysis uncovers the cell differentiation and functionalization for air breathing of frog lung"

Dear reviewers,

Thank you so much for your consideration on publishing our work. We have carefully revised our manuscript following your comments and suggestions.

Revision summary:

- Following your suggestions, we have removed the red blood cells from our single-cell data before conducting comparative analyses between stages and species.
- In the revised version, we have integrated the published scRNA-seq datasets from *Xenopus laevis*, mouse, and human lungs to enhance the reliability of our cell annotations using SAMap. Additionally, we have conducted cross-species analysis to elucidate the evolution of vertebrate lung cells.
- Following your suggestions, we now use Cell Rank v2 to perform pseudotime and trajectory analysis.
- All the comments have been revised or explained thoroughly. The manuscript has been edited by native English editors from the MDPI English editing service. The certificate is attached as a supplementary data.

Details about all our revisions are provided below. These modifications suggested by you clearly improved the quality of the manuscript. We hope you will agree that we have appropriately addressed all of your concerns and that the manuscript has been greatly improved as a result. Thanks again for your work!

Reviewers' comments:

Reviewer #1 (Remarks to the Author):

This manuscript reports ScRNAseq analysis of lung development in *Microhyla fissipes*, a land-dwelling frog species. The ScRNAseq analysis included lungs at 4 developmental stages, namely S41, S44, Sub-adult (post metamorphosis), and Adult, representative of the aquatic stage, early, middle, and late stages of terrestrial transition, respectively. Through ScRNAseq analysis, they identified the 4 major types of lung cells, which could be further divided into 21 subtypes of cells, and found that the proportions of 4 major cell types varied significantly among the 4 stages analyzed. They further analyzed stage-associated differences of some cell types and their featured differential expression of genes and transcriptional factors (TFs), and the potential developmental linkage of the cells through differentiation trajectory analysis over pseudo time. They found that the *Microhyla fissipes* lung comprised only one type of alveolar epithelial cells (AECs), contrasting with the facts that two types of AECs in the lungs of mice and other mammals, and the AECs exhibited temporal dynamic expression of feature genes and TFs with development. This a comprehensive manuscript that revealed how frog lung likely developed through cell type transformation during metamorphosis to adopt its habitual life change from aquatic to terrestrial environment. The manuscript is highly valuable for exploring the molecular mechanism underlining lung development in vertebrates. Below are some points for consideration

Reply: Thank you for investing your time in reviewing our work and provideing valuable comments and suggestions. All of your comments have been carefully addressed.

1. Through ScRNAseq analysis, only one type of AECs was identified (Fig. 1C), which expressed mouse AEC2 marker SFTPB (Fig. 4B and 5B-C), along with one type of AEC progenitors (Fig. 1C), which expressed the epithelial

progenitor marker gene SOX9 (Fig. 4B). How was this type of AEC progenitors compared with mouse AEC progenitor cells, e.g., any conservations among the featured differential expression of genes and/or TFs?

Reply: Thanks for pointing out this for us. After removing red blood cells from consideration, we repeated analyses of the single-cell data and annotated the cell type based on a cross-species alignment. No AEC progenitor cell cluster, corresponding that of mouse AEC progenitor, were identified in *M. fessipes* lungs. Since there is lack of cross-species evidences, we no longer concluded the cell cluster described in our old version of MS as AEC progenitor. The related result and discussion were removed from the revised version. Despite this regret, cross-species analysis has led us to new discoveries.

- 2. In the colocalization experiments (Fig. 4B), the FISH signal for SOX9 seemed to be very weak and need to be improved. Other than SOX9, did the authors try FISH with other epithelial progenitor marker genes to independently validate the AEC progenitor cluster?**

Reply: Due to the lack of cross-species evidence supporting the cell clusters as progenitors, we abandoned the FISH experiments of verifying it as a progenitor cell in the revised version. We also appreciate your rigorous assessment and insightful suggestions.

- 3. In Fig. 5B, the FISH indicated that AHNAK (one of the mouse AEC1 marker genes) colocalized with SFTPB (mouse AEC2 marker gene) very well, except for some small green dotted signals in the merged imagines. What were those green dotted signals representing?**

Reply: Thank you for your comments. Given that these small green dots are all regular circular shapes, we infer them to be fluorescence noise in fluorescence in situ hybridization (FISH). The reasons for the appearance of fluorescence noise in

FISH may include two factors: (1) Non-specific fluorescence: The dye itself may generate fluorescence rather than binding to the target DNA. Background fluorescence can result from impurities or chemical reactions. (2) Issues in fixation and processing steps: Problems during sample processing, such as incomplete fixation or excessive washing, may result in leakage or confusion of fluorescence signals. Regardless of the reasons for the appearance of these small green dots, they do not affect our conclusions.

In Fig. 5C, on the UMAPs, the 4 mouse AEC1 marker genes including AHNAK seemed not to co-express strongly in the AEC cluster (the AEC cluster were strongly red instead of yellow as expected). What is the explanation for this discrepancy?

Reply: Thanks very much for your valuable comments. On the UMAPs, the mouse AEC1 marker genes including AHNAK seemed not to co-express strongly in the AEC cluster. This discrepancy can include two reasons as follow: (1) The *AHNAK* expressed in matured AECs. We conducted the FISH of alveoli in the central of adult *M. fessipes* lung, the AECs distributed in this area basically are matured cell. The FISH signal is clear and convincing. Since the single cell atlases of *M. fessipes* lung captured AECs from various development stages and locations, various cell states likely expressed variable levels of AHNAK. AECs with low level of AHNAK were in a red instead of yellow color. (2) UMAP displays the expression profiles of different cells overlaid. We can see many yellow dots are obscured at the bottom layer. Therefore, we revised our conclusions: “matured *M. fessipes* AECs simultaneously expressed the markers of mouse AT1 and AT2”. Lines 201-204.

- 4. There were some inconsistencies between figures and figure legends: for example, Fig1 E and F mismatched with the corresponding figures; in Fig. 5, the figure legends regarding panels A to D didn't match the panels in the figure either (only A to C presented in the figure). The authors should also carefully**

check the citations of these figures in the text to make sure they were correctly referred in Results and Discussions.

Reply: Thanks for pointing out this for us. We have carefully checked the citations of these figures in the text throughout the manuscript.

- 5. In Fig. 3A as well as in the Supplemental Fig. 2C, 3A-B, 4B-C &4E, 5B &5D, there were colored highlight bars with different length in the GO terms panels. It is useful to define what different colors represent.**

Reply: Thanks for pointing out this for us. We have reorganized the figures in new version of MS. We defined what different colors represent in our figures.

- 6. There were some grammar errors in the text and figure legends should be corrected. The citations to the figures in the Results and Discussion should be carefully verified to correct mistakes, such as “Four distinct types of RBCs were identified (Figure 1A).” (line 309): Fig1A had nothing to do with RBC identification.**

Reply: Thank you for bringing this to our attention. We have revised these errors thoroughly. In addition, the manuscript has been edited by native English editors from the MDPI English editing service. The certificate is attached as a supplementary data.

Reviewer #2 (Remarks to the Author):

The work by Change et al. provides a comprehensive analysis of cell heterogeneity in amphibian lungs, with a specific focus on characterizing AECs (alveolar epithelial cells) and their developmental trajectories during the transition to air breathing. Leveraging the unique attributes of land-dwelling frogs, particularly *Microhyla fissipes*, the researchers performed single-cell RNA sequencing (scRNAseq) across four developmental stages to elucidate gene expression dynamics during the terrestrial transition. Through bioinformatics methods, the

authors unveiled the composition and heterogeneity of cell types throughout this transition, providing a crucial resource for advancing our understanding of lung development.

The study yields novel insights into the temporal dynamics of epithelial and endothelial cells during lung development, highlighting the existence of distinct cell states. Identification of transition cell states, exhibiting both proliferative and differentiating potentials, especially during critical transitional stages, suggests a strategic mechanism for rapid morphogenesis of the blood-air barrier during the metamorphic climax. The researchers emphasize the maintenance of growth, proliferation, and differentiation capabilities in epithelial and endothelial cells, orchestrated by specific transcription factors. This agrees with the activation of adaptive responses, such as surfactant proteins and oxidoreductases, signals preparation for the transition from a hypoxic to a hyperoxic environment. Delving into an evolutionary perspective, the study proposes that the diversity observed in AEC types and progenitors in amniotes reflects an advanced adaptation to terrestrial life.

Reply: Thank you for your positive comments on our manuscript.

While the technical foundation of the paper is robust, the authors are cautioned against the potential overinterpretation of results. This is particularly relevant given the paper's primary focus as a resource, though the authors are encouraged to provide further elaboration on result interpretation, especially in the Discussion section. Additionally, addressing the interpretation of cell states, both the proliferative and mature transitions, and exploring factors driving these transitions at bifurcation points would enhance the paper's depth.

Reply: Thanks very much for your valuable comments. We have updated the analysis methods and added a series of analyses to enhance the depth and significance of our study. For example, we have elucidated the differentiation trajectories and mechanisms of mesenchymal cells, identified five cell fates and their associated driver genes and

provided further elaboration on result interpretation, detailed below.

- (1) Matrix FB fate 1 serves as the origin of the MC lineage, characterized by the robust expression of genes related to the ribosomal protein subunit, indicating rapid cell growth and proliferation.
- (2) Matrix FB fate 2 is differentiated from Matrix FB fate 1 and it may act as a central hub for signaling, linking different cellular compartments throughout the lungs.
- (3) The FB fate is driven by the genes involved in the synthesis of collagen and extracellular matrix (ECM) components. The ECM, derived from FBs, not only provides structural support for pulmonary cells, but also plays a critical role in the regulation of developmental organogenesis and homeostasis.
- (4) SMC phenotype is characterized by the expression of genes encoding "contractile" proteins, and their transcriptional regulator *MYOCD* identified as its differentiation drivers.
- (5) The *WNT* signaling and *BMP* signaling pathways may coactivate pulmonary chondrocyte differentiation.

Minor comment: Please make sure all abbreviations are spelled out in the text.

Reply: We have revised it throughout the manuscript. Thanks for your valuable work.

Reviewer #3 (Remarks to the Author):

Chang et al. conducted a study on the cell atlas of the developing lung of a land-dwelling frog using single cell RNA-seq. They also performed comparative cellular analyses across different cell types and developmental stages, identifying cell types such as AECs and candidate genes (e.g., TFs) that may be involved in the lung development of land-dwelling frogs. While this manuscript presents an important resource for studying the evolution and development of vertebrate lungs, it lacks detailed analysis and robust results regarding these findings. Here are my main comments:

1. The overall cell classification is too simplistic. I suggest that the cell annotation

for the amphibian lung should reference the *Xenopus* Cell Landscape from Dr. Guo Guoji's laboratory, which can be found at <https://bis.zju.edu.cn/XCL/landscape.html>.

Reply: Thanks very much for your valuable comments. We have revised the manuscript carefully following your comments. These include:

(1) we reanalyzed the single-cell data after removing red blood cells. This improves the resolution of cell clustering and annotation.

(2) Cell clusters were annotated based on the expression levels of conservative cell markers from *Xenopus* Cell Landscape and Cell Marker 2.0 Database. Cell markers were visualized using “FeaturePlot” function (Figure 1C).

(3) To ensure the robustness of the cell annotations, we integrated the lung scRNA-seq datasets from *X. laevis*, mouse, and human into our analyses using SAMap (Figures 6). After calculating the alignment scores for each cell type across species, we annotated the lung cells of *M. fissipes* according to their correlations with those of *X. laevis*, mouse, and human lungs. This improved the robustness of cell annotation remarkably and enabled the following interspecies analyses.

Thanks for your valuable suggestion.

2. It is essential to assess the reproducibility of the cell proportions in each sample. I recommend increasing sample replication or employing alternative cell counting methods for validation. Neither fig.1D nor Supplementary Figure 1C demonstrate the consistency of cell proportions among different individuals at the same developmental stage. Additionally, the proportion of red blood cells appears to be quite high. The manuscript does not mention the treatment of red blood cells (e.g., cleaning, perfusion), which could introduce errors in comparing cell proportions across different development stages.

Reply: We tried to further support the single-cell data through quantitative histological analyses. Unfortunately, the limited cell types identified through histological identification is insufficient to support reliable quantitative analysis. To avoid

overreaching conclusions, we have opted to abandon the conclusions drawn from the analysis of cell proportions in the revised version. As for red blood cells, we attempted removal through centrifugation, but a significant number of red blood cells were still captured. In this manuscript revision, we removed the data associated with red blood cells through data analysis. In addition, we have added this process in Method section (Lines 448-452).

3. The abstract and methods sections indicate the use of scRNA-seq technology, but the figure labels mention snRNA-seq. This discrepancy should be clarified.

Reply: Thanks for pointing out this for us. It should be scRNA-seq technology. We apologize for our typing mistakes.

4. In addition, there are many spelling and punctuation errors in the manuscript, which requires a native english speaker to revise.

Reply: Thanks for pointing out this for us. We have revised these errors thoroughly. The manuscript has been edited by native English editors from the MDPI English editing service. The certificate is attached as a supplementary data.

REVIEWERS' COMMENTS:

Reviewer #1 (Remarks to the Author):

The changes and responses are satisfactory

Reviewer #2 (Remarks to the Author):

The authors have now addressed my concerns from the initial submission. I do not have any further comments.

Reviewer #3 (Remarks to the Author):

The author answered almost all my questions.

In the revised version, the author performed the cross-species comparison of cell types. The authors only used SAMap tool to do cross-species analysis. Considering the bias of this tool, the author needs to use multiple methods to validate the results of cross-species comparison. For example, the author mentioned in Line 223 "some cell types, such as the EC_Ls and ciliated cells, were not identified in *X. laevis*." Furthermore, the Epi cells shown in fig.6 are homologous in *Microhyla fissipes*, *Xenopus laevis* and human, but not in mice, which is puzzling.

Point-by-point response letter to Reviewers' comments

Manuscript number: COMMSBIO-23-3846A

Title: " Single cell RNA analysis uncovers the cell differentiation and functionalization for air breathing of frog lung"

Dear reviewers,

Thank you so much for your consideration on publishing our work. These modifications suggested by you clearly improved the quality of the manuscript. Thanks again for your work!

REVIEWERS' COMMENTS:

Reviewer #1 (Remarks to the Author):

The changes and responses are satisfactory

Reply: Thank you for your feedback. We are pleased to hear that the changes and responses were satisfactory. We appreciate your time and effort in reviewing our work.

Reviewer #2 (Remarks to the Author):

The authors have now addressed my concerns from the initial submission. I do not have any further comments.

Reply: Thank you for your review and for acknowledging that our concerns from the initial submission have been addressed. We appreciate your time and valuable feedback

Reviewer #3 (Remarks to the Author):

The author answered almost all my questions.

In the revised version, the author performed the cross-species comparison of cell types. The authors only used SAMap tool to do cross-species analysis. Considering the bias of this tool, the author needs to use multiple methods to validate the results of cross-species comparison. For example, the author mentioned in Line 223 "some cell types, such as the EC_Ls and ciliated cells, were not identified in X.

laevis. " Furthermore, the Epi cells shown in fig.6 are homologous in *Microhyla fissipes*, *Xenopus laevis* and human, but not in mice, which is puzzling.

Reply: Thanks very much for your valuable comments. Regarding the observed absence of certain cell types in specific lineages, we want to clarify that this result is not attributed to bias in the SAMap tool.

1. SAMAP is a robust cross-species single-cell transcriptome comparison analysis tool [1, 2].
2. Our analysis utilized well-established single-cell atlases for *Xenopus laevis*, mice, and human lung tissues, where cell clustering had been previously performed (Fig. 1-3). The absence of specific cell types is intrinsic to these atlases and not induced by SAMap analysis.
3. The absence of specific cell types may stem from genuine biological differences, experimental biases, or biases in cell clustering analysis. We have discussed these possibilities in Section Discussion, for example:

“Several cell types of *M. fissipes*, such as EC_Ls and ciliated cells, had no counterpart in *X. laevis*¹¹. This may be explained by the low proportions of lung cells in the single-cell atlas of *X. laevis*¹¹, leading to some cell types not being captured, as ciliated cells are evolutionarily conserved and have been identified in lungfish¹², *M. fissipes*, and mammal lungs⁹.”

“Interestingly, only one type of EC was identified in the lungfish¹² and *X. laevis* lungs¹¹, whereas more than one type of EC was identified in the *M. fissipes* and mammal lungs⁹. Further analysis revealed that *VWF*¹¹, the EC marker of *X. laevis*, was commonly expressed in the EC_Ls and EC_Vs of *M. fissipes*. And the EC_Ls and EC_Vs of *M. fissipes* exhibit strong correlations with the EC_Ls and EC_Vs of the mammals, respectively, underscoring the robustness of EC types in *M. fissipes* lungs. Thus, there may be two explanations for the difference in EC types between *X. laevis* and *M. fissipes* lungs. (1) *M. fissipes* has evolved two distinct EC types to adapt to terrestrial environments due to their distinct habitats, or (2) the limited discernment on the ECs of *X. laevis* single-cell atlases, which were constructed with multiple tissues, resulted in a relatively small number of lung cells”.

4. Due to the limitations of existing datasets, we cannot construct a perfect cross-species cell type mapping. Nevertheless, our data substantiate our conclusions. As single-cell datasets continue to improve incrementally, we anticipate achieving a more refined cross-species cell mapping for elucidating vertebrate cell evolution.

Thank you once again for your valuable advice. I have learned a lot from your insights.

Fig1. single-cell atlas of mouse

Fig2. single-cell atlas of human

Fig3. single-cell atlas of *X. laevis*

References

1. Chen DS, Sun J, Zhu JC, Ding XN, Lan TM, Wang XR, Wu WY, Ou ZH, Zhu LN, Ding PW, et al: **Single cell atlas for 11 non-model mammals, reptiles and birds.** *Nature Communications* 2021, **12**.
2. Zhang R, Liu Q, Pan S, Zhang Y, Qin Y, Du X, Yuan Z, Lu Y, Song Y, Zhang M, et al: **A single-cell atlas of West African lungfish respiratory system reveals evolutionary adaptations to**

terrestrialization. *Nature Communications* 2023, 14:5630.